# Better start to bilingual development: Bridging parental beliefs and science through early intervention for Polish families living in Norway

Agnieszka Dynak[1]*, Katarzyna Bajkowska[1,2], Jolanta Kilanowska[3], Joanna Kolak[4], Magdalena Krysztofiak[1], Magdalena Łuniewska[1], Karolina Muszyńska[1], Nina Gram Garmann[5], Ewa Haman[1]*

1 Faculty of Psychology, University of Warsaw, Warsaw, Poland, 2 Institute of Psychology, Polish Academy of Sciences, Warsaw, Poland, 3 Department of Primary and Secondary Teacher Education, Oslo Metropolitan University, Oslo, Norway, 4 Department of Psychology and Human Development, University College London, London, United Kingdom, 5 Department of Early Childhood Education, Oslo Metropolitan University, Oslo, Norway

* agnieszka.dynak@psych.uw.edu.pl (AD); ewa.haman@psych.uw.edu.pl (EH)

## Abstract

We present an intervention designed to align parental beliefs about language development and bilingualism with evidence-based recommendations. Its effectiveness was evaluated in a pre-registered randomised controlled trial. Seventy-six Polish parents living and intending to raise their children in Norway participated in either the intervention ($n = 40$) or a control condition ($n = 36$). We assessed the extent to which parental beliefs were aligned with scientific evidence on language and bilingual development before the intervention, immediately after the intervention, and when their children were about 9 months old. We found that the intervention was effective, i.e., it increased the alignment of parental beliefs with scientific evidence (preregistered ANOVA analysis 1). The benefits of the intervention were only seen in the experimental group. The gains were maintained when the children were around 9 months old (non-preregistered ANOVA analysis 2). A non-preregistered exploratory regression analysis 1 showed that participation in the experimental intervention accounted for 28.5% of the variance in parents' alignment of beliefs with science at posttest. Additional predictors of this alignment included participants' initial alignment at baseline, their level of engagement in the workshops, and their educational attainment. A preregistered exploratory regression analysis 2 showed that the greatest gains (the difference between posttest and pretest scores) from taking part in the intervention were found among those participants whose initial beliefs were the least aligned with science (at pretest). The gains were not associated with participants' individual engagement in the intervention, prior parenting experience (raising an older child), or their level of education. Future directions based on these data will include investigating whether changes in parental beliefs translate into improved language outcomes for their children.

**Data availability statement:** The data and script along with the preregistration are available at OSF (https://osf.io/7a2zp). The translations of the intervention content and materials to English and Norwegian along with the Polish original are publicly available on the OSF (https://osf.io/46fvq/overview).

**Funding:** The study has been funded by the Norwegian Financial Mechanism (https://eeagrants.org/en/fmo) for 2014-2021 via NCN GRIEG programme (Project. no. 2019/34/H/HS6/00615) awarded to EH, and partially funded by the Faculty of Psychology, University of Warsaw (https://psych.uw.edu.pl/, no. 501-D125-01-1250000/5011000230 awarded to AD). During some stages of the preparation of this paper AD's work was financed by the Polish Ministry of Science and Higher Education grant IDUB UW Action (https://inicjatywadoskonalosci.uw.edu.pl/en/, grant no. 501-D125-20-4004310). The funders had no role in study design, data collection and analysis, decision to publish, or preparation of the manuscript.

**Competing interests:** The authors have declared that no competing interests exist.

## Introduction

### The importance of language input in supporting children's language development

High quality interactions with caregivers are crucial for language development [1] both in monolingual and bilingual context [2,3], as children's language skills in a given language have been associated with the amount of exposure to that language [4–6]. Parents are the primary source of language input for their children from birth. However, families vary substantially in the way they talk to their children. For instance, parents of low socioeconomic status (SES) may engage in fewer verbal interactions with their children compared to more affluent counterparts, as it was first described in a seminal study by Hart and Risley [7]. This gap, known as the "30 Million Word Gap", suggests that by the age of three, children from low SES families may hear up to 30 million fewer words than their high SES peers. This disparity is associated with later differences in vocabulary, IQ scores, and academic achievement [8]. Although Hart and Risley's original study has been criticised and the vocabulary gap may be smaller than estimated [9], other studies have also shown significant differences in the language environments of children from differing SES backgrounds [e.g., 10–12]. The disparity is not only in the quantity but also in the quality of child-directed speech. Parents from low SES backgrounds tend to use less diverse vocabulary, shorter sentences, and more directive (less eliciting) language [13]. All of these factors affect children's language outcomes, and the effect of language quality may outweigh the effect of language quantity [14].

One of the important factors that has been shown to mediate between SES and children's language development is parental knowledge of how children develop [13] and their attitudes towards various communication and caregiving behaviours [15]. In the context of multilingualism, these behaviours also depend on parental knowledge and beliefs about multilingualism and multilingual development, which in turn influence children's development in all their languages [16].

### Parental behaviours that support children's language development

Several parental behaviours have been shown to support a child's language development. As mentioned above, the quantity and quality of child-directed speech are two of the most important aspects of parental language support. Moreover, the quantity and quality of language heard by a child are highly correlated [17] so it is challenging to disentangle their influence on language outcomes. However, the amount of language a child hears is likely to be more important in the early stages of development, while language richness plays a bigger role in later stages [18].

The properties of language quality, such as higher lexical diversity or higher sentence complexity, are predictive of a child's development in both monolinguals and multilinguals. For example, parental responsiveness, i.e., providing prompt, contingent, and appropriate responses to the child's non-verbal signals, has been shown to facilitate early language development [19]. Engaging in frequent conversational turns with children was shown to be more strongly associated with their vocabulary growth

[20] or general verbal skills [1] than the quantity of input. The frequency of conversational turns is also associated with both structural [21], and functional [1] differences within the neuronal language circuit in the child's brain. Similarly, the use of infant-directed speech (also known as *baby talk* or *parentese*) can enhance young children's language development, in particular their word production [22]. This effect is independent of the language used [23].

On the other hand, some practices may hinder language development. For example, the quality and quantity of a linguistic environment may be reduced by the amount of parent and child media use. The presence of media in the background, frequent parental use of media during the child's routines, and the child's use of media can decrease the quantity and quality of parent-child interactions and negatively impact the child's language skills [24–26].

There is a substantial body of research evaluating parental interventions that include a language component [see [27] for meta-analysis]. Some of them directly measured the changes in parental knowledge or beliefs due to taking part in interventions. All of these studies (internet version of PALS [28]; Thirty Million Words-Well Baby intervention [29]; 3Ts – Tune In, Talk More, Take Turns – Home Visiting [15,30]) showed that taking part in the interventions resulted in the increase of parenting knowledge [28] or knowledge about early childhood cognitive and language development [15,29,30]. In most cases ([15,30] and in high intensity training described by List and collaborators [29]), the increase in knowledge was also related to enriched parent-child interactions.

Although we do know a lot about what facilitates and what hinders children's language acquisition, the challenge for parents of bilingual and multilingual children is bigger than for those of monolingual children. As the amount of children's hours awake is limited, it can be difficult to provide language support of high quality and quantity in more than one language. On average, multilingual children have limited exposure to each of their languages compared to their monolingual peers [31] due to the time constraints.

Research on the relation between certain parental behaviours and children's language development has resulted in recommendations for caregivers [32] and strategies that can be implemented to promote language development in children, e.g., *Support, Ask, Expand* by Boyce et al. [33], *1,2,3, Tell Me What You See*, by Brannon and Dauksas [34], and *Comment, Ask and Respond* by Cole et al. [35]. Such recommendations and strategies may be taught through parent-oriented interventions, which can effectively enhance children's language development, in particular their expressive language [27]. Studies have shown that parental knowledge about language development can successfully be modified [e.g., [29,36] and as a result, the quality and quantity of parental language may be improved [see 37 for the review].

Up to date, there are few interventions tailored specifically for bilingual families, providing support that directly addresses the challenges associated with raising children in a multilingual environment and supporting their home language development [see 38 for the review; see also 28,29]. One such intervention that has proven to enhance both parental responsiveness and children's outcomes, including early language skills, is *Parents As Literacy Supporters (PALS)* [32]. Anderson and collaborators [39] provided a version of this programme for bilingual families of preschool children from diverse SES backgrounds in Canada. Participation in the program entailed over 20 on-site group sessions for families, focusing on supporting literacy skills and related topics, such as developing mathematical skills. Upon completion of the programme, children exhibited greater improvements in early literacy skills than what would be expected solely through natural development and preschool education. However, the evaluation process did not include typical language measures, and there was no control group (involved in a non-related intervention) for comparison. PALS was later adapted to various settings. For example, Feil and collaborators [28] adapted the program to be internet-based and provided it to a diverse group of participants (38.7% of whom were Hispanic mothers). The study included a control group and showed that mothers who participated in PALS not only demonstrated a significantly greater increase in parenting knowledge but also improved language-supportive behaviors. Children's language outcomes were also improved compared to those in the control group, who received attention-control training.

While the studies reviewed here target mostly families with children above 3 years of age, Adamson et al. [40] suggest that interventions aiming at supporting parents in fostering high-quality interactions with their children should be provided

before the child starts to speak. Research on parental language interventions in low-income families has shown that the interventions are most effective when they start early, either prenatally or within the first year of the child's life [41]. Nevertheless, most interventions currently target families with children who have already exhibited language delays, language disorders, or communication-related disabilities such as autism spectrum disorder [42]), or they target families with low-SES backgrounds, and little is known about the effects of language interventions for children outside these groups.

Offering early interventions to parents of bilingual children (i.e., specific groups of children on a population level) could enhance the children's development, enabling them to thrive even further. Most importantly, timely support in each of the two languages would benefit children who indeed prove to have additional needs. Researchers reviewing existing studies on the subject report a need for population-level interventions [43], which – if implemented early enough – could have a preventive role.

To our knowledge, there are no studies to date on the effectiveness of language interventions addressed to parents of bilingual infants or toddlers (regardless of the SES level). Studies on interventions targeting parents of older children **–** preschoolers and school-aged children **–** are often limited to those on dialogic reading [44] and are not tailored specifically to the needs of bilingual families [45]. Delivering this type of intervention to bilingual families was proven to enhance the quality of home language and literacy environment, which in turn improved children's language outcomes (e.g., narrative skills, [33] or the pace of learning new vocabulary, [34]). A meta-analysis by Heidlage et al. [27] has shown that interventions that modified parental language during play and/or everyday routines yielded larger effect sizes compared to those focusing solely on shared book reading. Our aim was to design an intervention tailored to the specific needs of bilingual families, applicable to a variety of interactional contexts.

## The current study

We present an intervention addressed to Polish-speaking parents living in Norway and expecting a child. In Norway, systematic advice is provided via the health care system on how to support language development. However it does not meet the complex needs of bi- and multilingual families. Children under the age of 5 and their parents are included in the infant health care programme [46]. Health care centres with public health nurses and a GP run this health-promoting, multidisciplinary service which consists of home visits, health check-ups, vaccination, health information, and other advice. Topics discussed with parents include the importance of interaction between parents and children, sleep, nutrition, and the children's general development. Examples of parental practices to support the child's language development may be discussed during the check-up meetings. However, language development as a specific topic is addressed for the first time during the 8-month check-up [46], hence systematic support from birth may depend on an individual approach of a public health nurse.

This study aims to assess the effectiveness of the intervention in aligning parental beliefs about language development and bilingualism with scientific evidence on factors supporting language development. We also investigate whether the effect is long-lasting and what factors influence the alignment of participants' beliefs with science after the intervention. Our study was designed as a randomised control trial with two groups: intervention, focused on language development, and an active control group. The control group was provided with evidence-based workshops on sleep patterns in infants. Workshops in both groups were as similar as possible in terms of timing, structure, and communication with participants.

We hypothesised that the alignment of parental beliefs with scientific evidence on language development would increase significantly in the group that received the language intervention compared to the control group. We expected several factors to predict the degree to which participants' beliefs aligned with scientific evidence after participating in the experimental or control intervention. Specifically, we anticipated that participants in the experimental condition would show greater alignment with scientific evidence than those in the control condition, and that higher individual engagement in the workshop would be associated with stronger alignment at posttest. Based on previous research (e.g., [47]), we also expected demographic characteristics—such as educational attainment and prior parenting experience—to be associated with beliefs more closely aligned with scientific evidence.

We also investigate whether the effect is long-lasting and what factors influence participants' gains from the intervention. To the best of our knowledge, no studies to date investigated parental characteristics that bring larger gains (i.e., increased alignment with scientific evidence) from parental language intervention. We expect that a lower level of knowledge about language development and bilingualism at the beginning of the study will be negatively correlated with an increase in the alignment of those beliefs with scientific evidence after the intervention (which we call below "the gain"). We also expect that participants with lower education will gain more from the intervention than participants with higher education, as we expect participants with higher education to already know more about childcare and language development, compared to participants with lower education [47].

This study was preregistered at the stage of data collection before accessing the data. The preregistration is available on Open Science Framework (OSF: https://osf.io/7a2zp). Here we present both the analyses specified in the preregistration and additional exploratory findings. While the analysis of predictors of participants' gains from the experimental intervention (referred to as preregistered regression analysis 2) was preregistered as exploratory, we also report an additional exploratory analysis (referred to as non-preregistered regression analysis 1) investigating predictors of post-intervention alignment of beliefs with scientific evidence across both the experimental and control conditions.

## Methods

### Recruitment

We targeted parents born in Poland who were living in Norway and planning to remain there long-term. Eligible participants were either expectant parents or parents of infants aged ≤ 3 months. Recruitment was conducted online and by distributing materials (posters and leaflets) in healthcare centres, birth classes, and Polish communities (e.g., Polish parishes) in large cities in Norway (mostly Oslo). Online recruitment included posting advertisements on social media (e.g., Facebook groups) and websites targeted at Poles in Norway. Parents received detailed information about the study before they agreed to participate, without an indication of what exactly would be the topic of the workshop.

In 2023, there were 913 children aged 0–12 months of Polish parents living in Norway [personal communication, 48]. Due to the relatively small annual count of children with at least one Polish parent born in Norway, our recruitment was continuous through 11 months to maximise the number of participants.

### Participants

**Expected sample size.** To establish the targeted number of participants, we used the G*Power software [49] to conduct a series of power analyses. First, we checked what sample size was needed to obtain $1 - \beta = 0.8$ power at the standard $\alpha = 0.05$ error probability, when the medium effect ($d = 0.5$) was assumed. The number of participants required was $N = 102$ ($n = 51$ in each group) in between group comparisons. However, since the study was targeted at a small population (expectant parents living in Norway and speaking Polish), we concluded that recruiting such a large sample of participants was unlikely. Thus, we ran a second power analysis, with a large expected effect size ($d = 0.8$) and the required number of participants was $N = 42$ ($n = 21$ in each group). Since we expected an effect size to be between medium and large, we aimed for a number of participants in between the two numbers (42–102).

**The group recruited for the study.** We initially recruited 139 families, who were semi-randomly assigned into two groups: one experimental and one control group (receiving an intervention on sleep patterns in infants). We used a method of randomisation that minimises the imbalance in the levels of the covariates, with the following variables as covariates: (a) mother's education (primary school/ middle school – vocational/ secondary school/ incomplete higher education/ tertiary education – BSc; BA; Eng.; MSc; MA/ tertiary education – PhD), (b) child's expected sex (male/ female/ not yet revealed), (c) prior parenting experience (having older children in the household: yes/no). At the beginning of the study, participants who registered to the study were paired on the levels of covariates and assigned to the groups using simple randomisation. As some registered participants withdrew before the intervention started, there was an imbalance

in the levels of the covariates, thus we decided to further assign participants to the groups pseudo-randomly in a way that would minimise the imbalance. If two potential participants had the same levels of covariants, we used simple randomisation to assign them to groups.

Between January and November 2022, we delivered six rounds of workshops, with each round involving one experimental and one control workshop. Each workshop consisted of 3 meetings, each lasting 1.5 hours (4.5 hours in total), delivered online on the Zoom platform. Towards the end of the project, efforts were made to recruit participants for a face-to-face workshop edition, aiming to compare its effectiveness with the online workshop. Despite extensive efforts, only 8 participants were recruited, and they were assigned to an additional experimental group (language workshop), which made it unfeasible to proceed with the planned comparison between face-to-face and online formats. As a result, we decided to offer these participants the same online workshop as those in the randomized trial. These participants were not included in the main analyses testing the effectiveness of the intervention, as they were not randomly assigned to groups. They enrolled in the study expecting to participate in the workshop related to child language (as opposed to the randomised participants who were informed that they would take part in a workshop about supporting early child development in general). However, ultimately, they received the exact same experimental (language) intervention – delivered in the same format – as the participants in the randomised trial, and they completed the same set of questionnaires. For this reason, to enhance statistical power, data from these participants were included in the exploratory analyses. This encompassed both the examination of factors associated with post-intervention alignment of participants' beliefs with scientific evidence (regardless of intervention condition) and the analysis of predictors of the extent of benefit derived from participation in the experimental intervention. This – non randomised – subgroup was not originally planned and was therefore not included in the preregistration. The subgroup is referred to in the article as a non-randomised group.

We provided the intervention (experimental or control) to a total of 13 groups, comprising 95 participants. If the participant missed any of the intervention sessions, they were invited to take part in the corresponding meeting with the next available group. If they could not join the other group, they were given materials (electronic handouts) to study on their own.

Prior to each intervention, the participants were asked to complete the *Questionnaire on Beliefs about Language Development and Bilingualism* [50] which was created specifically for this study. If a participant did not complete the beliefs questionnaire before the intervention (time point 1, TP1) or after the intervention (TP2), their data were excluded from the analyses.

Out of the 95 participants who attended at least one of the three intervention sessions synchronously (session attendance is summarised in Table 1), 84 completed both the pretest and posttest measures and were therefore included in the current analyses. The final randomised sample consisted of $N=76$ participants ($n=40$ in the experimental group and $n=36$ in the control group). An additional $N=8$ non-randomised participants were included only in the exploratory analyses (thus $N=84$ in total). All participants who did not attend all intervention sessions were given the opportunity to review the material independently using PDF handouts.

In the preregistered analysis assessing the intervention's direct—i.e., short-term—effectiveness of the intervention (below: ANOVA analysis 1), we used data from participants attending the initial 12 intervention groups ($N=76$; $n=40$ in

**Table 1. Intervention session attendance by study group (N = 84).**

| Number of sessions attended synchronously | Randomised | | Non-randomised – language workshop | Total |
|---|---|---|---|---|
| | Experimental – language workshop | Control – sleep workshop | | |
| Three (all) | 24 | 24 | 7 | 55 |
| Two (one missed) | 11 | 8 | 0 | 19 |
| One (Two missed) | 5 | 4 | 1 | 10 |

experimental, and $n = 36$ in control group, randomised). In the exploratory non-preregistered regression analysis 1 investigating factors influencing the alignment of parental beliefs with scientific evidence after the intervention, we included data from the participants who took part in the additional language workshop as well ($n = 8$). Because of the missing data one of the participants had to be excluded from this analysis. This resulted in $N = 83$ participants in the exploratory analyses (see Table 2). In the exploratory preregistered regression analysis 2 investigating factors influencing the degree of the increase in alignment of parental beliefs with scientific evidence ('the gain'), we included the data from participants who took part in the additional language workshop as well ($n = 8$). This resulted in $n = 48$ participants in the preregistered regression analysis 2 (see Table 2).

In the analysis regarding the long-term effects of the intervention (maintenance of the changes in parental beliefs, below: non-preregistered ANOVA analysis 2), we included only data from the participants of the randomised groups who completed the beliefs questionnaire at all three time points: before the intervention (TP1), after the intervention (TP2), and when the children were 9 months-old (TP3). The drop-out rate at TP3 was substantial, so the sample in this analysis was much smaller ($n = 42$; $n = 17$ in the experimental group, and $n = 25$ in the control group), compared to the analysis assessing the immediate intervention effect (which included only TP1 and TP2 data).

## Data collection procedures

This study is part of the PolkaNorski Project (Project. No. 2019/34/H/HS6/00615) funded by the Norwegian Financial Mechanism 2014–2021 within the NCN GRIEG programme. The funders had no role in study design, data collection and analysis, decision to publish, or preparation of the manuscript.

Upon registering for the study (at TP1), parents were asked to provide demographic information through an online questionnaire. They were also given the online beliefs questionnaire [50]. The completion of these two questionnaires

**Table 2. Characteristics of participants included in the analyses of the effectiveness of the intervention (ANOVA analysis 1) and factors influencing the gain from the intervention (regression analysis 2).**

| Baseline characteristic | Randomised | | Non-randomised – language workshop | Total |
|---|---|---|---|---|
| | Experimental – language workshop | Control – sleep workshop | | |
| *n* | 40 | 36 | 8 | 84 |
| **Participant's gender** | | | | |
| Female | 37 | 31 | 7 | 75 |
| Male | 3 | 5 | 1 | 9 |
| **Prior parenting experience** | | | | |
| Expecting first child | 26 | 27 | 6 | 59 |
| Rising older child(ren) | 14 | 9 | 2 | 25 |
| **Highest educational level** | | | | |
| Secondary school | 3 | 4 | 1 | 8 |
| Incomplete higher education | 8 | 5 | 1 | 14 |
| Tertiary education – BSc; BA; Eng.; MSc; MA | 28 | 26 | 6 | 60 |
| Tertiary education – PhD | 1 | 1 | 0 | 2 |
| **Child's expected sex** | | | | |
| Male | 16 | 13 | 2 | 31 |
| Female | 13 | 13 | 1 | 27 |
| Not known | 11 | 10 | 5 | 26 |

**Note.** The mean age of all participants was 33.48 years ($SD = 4.21$). The age of randomised participants did not vary between conditions.

typically took place 0–6 days before the first meeting. Parents were then asked to complete the belief questionnaire twice more: After the end of the intervention (TP2), and 9 months after the child's birth (TP3). All the questionnaires were set up on the Qualtrics platform.

All parents also filled in a questionnaire on children's sleep patterns, as this was the theme for the control group intervention. The questionnaire was designed and pilot-tested in a similar way to the language questionnaire. Here, we will only report on the scores for the language questionnaires.

In addition to the questionnaire data, which is our primary focus here, a trained observer recorded the frequency of spontaneous verbal contributions made by each participant during the workshop. We use this data as an indicator of participant individual engagement in the workshop.

Based on an agreement with OsloMet, Norway, Data Protection Services (SIKT) has assessed that the processing of personal data in this project is in accordance with data protection legislation (reference number 551928). In Poland, the ethical committee at the Faculty of Psychology at the University of Warsaw has evaluated and approved of the research plans for PolkaNorski and for this study in particular (reference number 09/06/2021), including the processing of personal data.

## Intervention: Content and structure

The groups received an online workshop either concerning child language development and the role of parental behaviours for language development (experimental condition), or practices leading to creating healthy sleep and sleep behaviours in infants (control condition). The workshops were held in Polish (i.e., the first language of all the participants). The intervention programs follow the guidelines for effective language interventions and cover all the strategies recommended by Biel et al. [51]: building upon background knowledge, explicitly describing and illustrating content, providing opportunities to actively apply and generalise learned content in real-world contexts, and supporting metacognition (e.g., reflection and self-monitoring) throughout the training process.

Both the experimental and control intervention programs consisted of 6 blocks, each covering one topic. In the language (experimental) condition, the topics included: preverbal communication, milestones in early language development, quantity and quality of language input, infant-directed speech, bilingualism, and the impact of media on early language development. Each block involved a brief presentation of research-based facts on a specific topic, followed by practical suggestions for applying this knowledge when interacting with a child. This theoretical part was followed by an exercise to support practical skills. Some exercises were conducted with the whole group and some in smaller subgroups or pairs. After each exercise, participants were encouraged to share their experiences (e.g., discuss the specific elements that proved difficult for them). A research team member observed the workshops to ensure that the facilitator adhered to the script consistently across the groups, maintaining comparability in the delivery of the intervention.

Following each workshop, participants were provided with supplementary written materials. These materials included presentations from the workshop accompanied by a bibliography, resources on dialogic reading, a comprehensive description of language development, milestones during the early years of life, and practical suggestions on how to engage in quality play with a child to enhance language input. All the materials are publicly available on the OSF (https://osf.io/46fvq). The resources are provided in Polish (the original language of the intervention) together with English and Norwegian translations.

The control group took part in a workshop aimed at supporting healthy sleep patterns in infants. The program of this workshop was prepared by an expert in the area of infant sleep in collaboration with the research team. It was designed similarly to the language workshop (the same length, number of topics/ blocks, the same proportion of different types of activities, and similar style of exercises). Participants of this workshop were also sent supplementary materials after the workshop. Both workshops were run by a facilitator (three different people) with expertise in working with parents of infants. Each facilitator provided the workshop to both the experimental and the control group to eliminate the effect

of the facilitator. An independent observer monitored the interventions to ensure consistency between each round of interventions.

## Variables and measures

The dependent variable was the alignment of parental beliefs with scientific evidence on language and bilingual development, as well as on behaviours that support it. It was assessed with a 31-item criterion-reference *Questionnaire on Beliefs about Language Development and Bilingualism* [50]. Participants responded to all questions using a 5-point Likert scale (from "I totally disagree" to "I totally agree"). Responses aligned with scientific evidence, such as "I agree" or "I totally agree" (if the sentence was evidence-based) were deemed correct. An answer of "I have no opinion on that" was awarded 1 point, while answers not grounded in scientific evidence received 0 points. This scoring system yielded a total score ranging from 0 to 62 points. Sample items include: "Speaking to infants in a specific way (e.g., in a higher tone of voice, with exaggerated intonation, using diminutives) helps them learn to speak," (correct answer: agree; 2 points given for selecting "I agree" or "I totally agree") and "Leaving the TV/radio on in the background is a great way to give 0- to 2-year-olds extra chances to learn new words." (correct answer: disagree; 2 points given for selecting "I disagree" or "I totally disagree"). The questionnaire has good internal consistency (Cronbach's alpha = .83), based on the pilot study conducted on a larger number of items (*n* = 98) [52,53].

In the regression analyses, five predictors were taken into account: (a) the type of intervention that participant was assigned to (this measure was used as predictor only in non-preregistered regression analysis 1) (b) the level of scientifically based beliefs at study entry (the points that the participant achieved in the beliefs' questionnaire at TP1) (c) the level of individual engagement in the intervention (measured as the frequency of spontaneous verbal contributions made by each participant during the workshop) (d) the level of the participant's education (primary school/ middle school – vocational/ secondary school/ incomplete higher education/ tertiary education – BSc; BA; Eng.; MSc; MA/ tertiary education – PhD), and (e) prior parenting experience (raising an older child: yes/no). The second of those variables, parental beliefs at the study entry, was assessed using the same questionnaire as the outcome variable [50]. The third variable (participant's engagement in the intervention) was measured by the trained observer as the number of times that the participant spontaneously spoke during the workshop, e.g., shared their experience or asked a clarifying question. Whenever the participant spoke, the observer noted this fact in the observation sheet. The participant's words were not noted, just the simple fact of speaking. Participants were informed about the presence of the observer and her task before the workshops began. The final value of the engagement variable for each participant was calculated as the sum of the number of verbal contributions across all the sessions combined. In this way, the engagement variable reflected whether the participant attended each session (as missing a session precluded obtaining any activity points for that meeting) as well as whether their participation went beyond passive listening. The last two predictors (education and prior parenting experience) were collected in the demographic questionnaire.

## Analytic strategy

The article presents four analyses: (a) The mixed ANOVA analysis 1 testing the effect of the intervention that was preregistered (https://osf.io/kfgba); two regression analyses: (b) non-preregistered regression analysis 1, investigating predictors of post-intervention alignment of participant's beliefs with scientific evidence across both the experimental and control conditions and (c) preregistered regression analysis 2, identifying predictors of increase in the alignment between participants' beliefs and scientific evidence ('the gain'); (d) The fourth analysis – a mixed ANOVA analysis 2 examining whether the effects of the intervention were sustained over time. The fourth analysis was not included in the preregistration because it had not yet been planned at that stage.

As planned in the preregistration, we conducted a mixed ANOVA to test **the effectiveness of the intervention.** The score on the beliefs questionnaire [50] was a dependent variable, while group and time point were the independent

variables. Group assignment (experimental or control) was treated as a between-subject variable and time point (pretest – TP1 vs. posttest – TP2) was a within-subject variable. As the groups did not differ on the levels of control variables (measured by the two sample t-tests: one for the education level: $t(71) = 0.05$, ns; and one for the prior parenting experience: $\chi\text{-}squared = 0.486$, ns) there was no need to include any co-variates to the model.

In the **non-preregistered regression analysis 1,** linear regression was conducted to examine what were **the best predictors of the alignment of participants' beliefs with scientific evidence after taking part in the intervention (experimental or control)**. The outcome variable was the score on the beliefs questionnaire [50] in the posttest (TP2). The predictors were: (a) the type of intervention the participant was assigned to (experimental or control); (b) the level of scientifically based beliefs at study entry (at pretest – TP1); (c) the level of individual engagement in the intervention; (d) the level of the participant's education, and (e) whether there were already children raised in the household (yes/ no). We used hierarchical regression, introducing the predictors one by one in the order presented above. First, we entered group assignment, as our aim was to determine the proportion of variance in the outcome variable explained by participation in the experimental intervention. Next, we added the pretest score, given our expectation that posttest performance would strongly depend on baseline levels. We then introduced the engagement variable, followed by the two demographic factors – educational attainment and prior parenting experience.

To further investigate the relationship between the predictors and the level of parental beliefs alignment with scientific evidence, we first examined bivariate correlations between each predictor and the initial level of parental beliefs alignment (TP1 score) on the whole sample ($N = 84$). This allowed us to assess the extent to which the predictors were related to the baseline values of beliefs alignment at TP1. The correlations are considered supplementary to the primary regression analyses and were not specified in the preregistration.

In the **preregistered regression analysis 2,** linear regression was conducted to verify what were **the best predictors of the increase in the alignment ('the gain') of participants' beliefs with scientific evidence**. The outcome variable was the increase in the score on the beliefs questionnaire [50] between pretest (TP1) and posttest (TP2). This increase was calculated as the difference of scores between TP2 and TP1 scores. The predictors were: a) the level of scientifically based beliefs at study entry (TP1 score); b) prior parenting experience; c) the level of individual engagement in the intervention; and d) the level of each participant's education. We used hierarchical regression, introducing the predictors one by one in the order presented above.

To further investigate the relationship between the predictors and the increase in parental beliefs alignment with scientific evidence, we first examined bivariate correlations between each predictor and the initial level of parental beliefs alignment (TP1 score) on the subgroup of participants that participated in the experimental intervention ($N = 48$). This allowed us to assess the extent to which the predictors were related to the baseline values. The correlations are considered supplementary to the primary regression analyses and, as such, were not specified in the preregistration.

To test **if the effect of the intervention persisted over time**, we conducted a mixed ANOVA with the score on the beliefs questionnaire [50] as a dependent variable and group and time point as independent variables. This analysis was not included in the preregistration. Group assignment (experimental or control) was treated as a between-subject variable and time point (pretest – TP1 vs. posttest – TP2 vs. long-term maintenance test – TP3) was a within-subject variable. As the groups did not differ on the levels of control variables (measured by the two sample t-tests: one for the education level: $t(29) = -1.22$, ns; for the prior parenting experience: $\chi\text{-}squared = 1.494$, ns) there was no need to include any co-variates to the model.

All the data were analysed in R Statistical Software (v4.4.0; R Core Team, 2024). The data and the script are available at OSF (https://osf.io/kfgba).

## Results

### Effectiveness of the intervention: Preregistered ANOVA analysis 1

A mixed ANOVA on the data obtained from participants who were assigned to groups randomly and filled in both the pretest and posttest questionnaire ($N = 76$, $n = 40$ in the experimental group, and $n = 36$ in the control group) was

conducted to test the effect of the intervention on the alignment of parental beliefs with scientific evidence. The effect of the between-subject variable: group (the type of the workshop: experimental or control) on parental beliefs was significant ($F(1,74) = 13.56$, $p < .001$, generalized $\eta^2 = .13$), indicating that the beliefs of parents from the experimental group were more aligned with scientific evidence than the beliefs of parents from the control group and that the effect size was medium (see Table 3 for descriptive statistics of parental beliefs about language development and bilingualism across groups and timepoints and Table 4 for a summary of the model results). Bonferroni post-hoc tests revealed that the groups did not differ significantly in the level of the alignment of beliefs with scientific evidence at the pretest but at posttest the beliefs in the experimental group were significantly more aligned with scientific evidence than in the control group (mean difference = 9.4, $p < .001$). The effect of the within-subject variable: time (pretest vs posttest) on parental beliefs was also significant ($F(1,74) = 31.79$, $p < .001$, generalized $\eta^2 = .08$), indicating an increase in the alignment of parental beliefs with scientific evidence after the intervention and that the effect size was medium. Bonferroni post-hoc tests revealed that the parental beliefs were significantly more aligned with scientific evidence after the intervention in the experimental group (mean difference = 7.9, $p < .001$) but not in the control group. The interaction effect between group and time on parental beliefs was significant as well ($F(1,74) = 11.77$, $p < .001$, generalized $\eta^2 = .03$, small effect size) – the increase between the two timepoints in parental beliefs' alignment was steeper for the experimental group than for the control group (see Fig 1). See Table 4 for a summary of the mixed ANOVA analysis 1 and Table 5 for the Bonferroni post-hoc tests examining intervention effectiveness.

**Predictors of parental beliefs' alignment with science following participation in the experimental or control intervention: Non-preregistered regression analysis 1**

A hierarchical linear regression was conducted to identify predictors of post-intervention alignment of parental beliefs with scientific evidence. The outcome variable was the beliefs questionnaire score at posttest (TP2 score). Predictors were entered sequentially in the following order: (1) group assignment (experimental vs. control), (2) baseline score at pretest (TP1), (3) engagement during the workshops; followed by the demographic variables: (4) education level and (5) prior parenting experience (having older children in the household: yes/no) (see Table 6 for detailed statistics for each model). Because the engagement value was missing for one participant, analyses were conducted on $N = 83$ complete cases.

Step 1 showed a significant effect of group, indicating higher TP2 alignment in the experimental group, and explaining 28.5% of variance in TP2 scores ($R^2 = .29$, adj. $R^2 = .28$), $F(1,81) = 32.34$, $p < .001$. Adding baseline alignment (TP1 score)

**Table 3. Descriptive statistics for questionnaire scores on parental beliefs about language development and bilingualism across groups and timepoints (TP1, TP2).**

| Group | Pretest (TP1) | | Posttest (TP2) | |
|---|---|---|---|---|
| | *M* | *SD* | *M* | *SD* |
| Experimental – language workshop n = 40 | 44.3 | 9.03 | 52.2 | 6.82 |
| Control – sleep workshop n = 36 | 40.9 | 9.05 | 42.8 | 8.76 |

**Table 4. Preregistered mixed ANOVA Analysis 1: Results on intervention effectiveness.**

| Effect | F | df1 | df2 | *p* | η2 |
|---|---|---|---|---|---|
| **Group** | 13.56 | 1 | 74 | <.001 | .13 |
| **Time point** | 31.79 | 1 | 74 | <.001 | .08 |
| **Group x Time point** | 11.77 | 1 | 74 | <.001 | .03 |

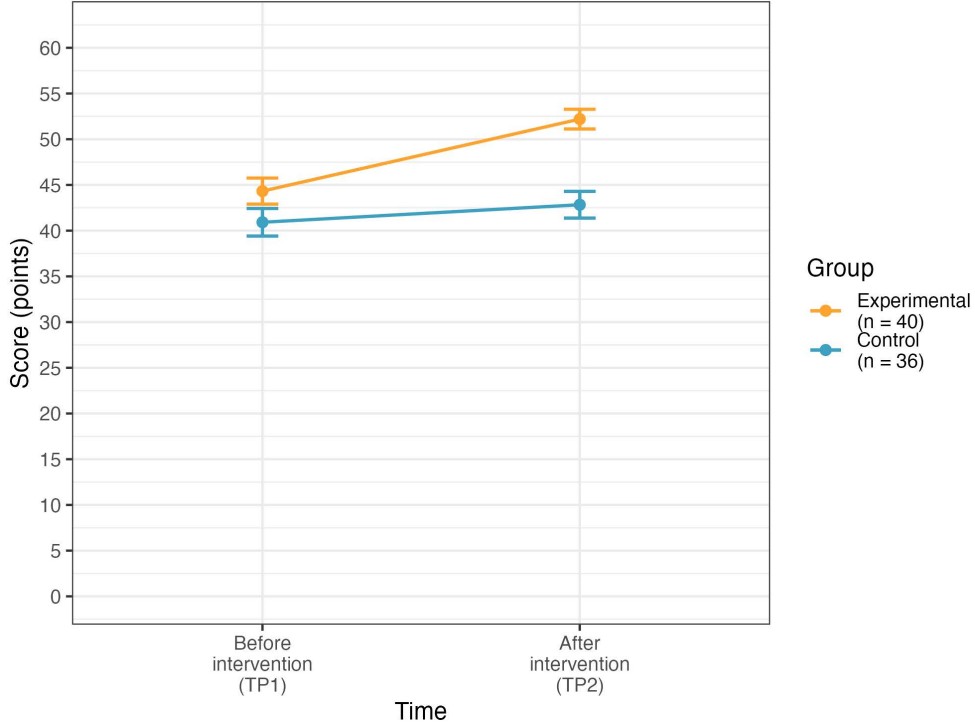

**Fig 1. Participants' performance at pretest and posttest in the experimental and control groups.** Error bars represent standard errors of the mean (±SE).

**Table 5. Bonferroni post-hoc test results on intervention effectiveness.**

| Group comparison | Mean difference | *p* |
| --- | --- | --- |
| TP1: Experimental vs control | 3.2 | ns |
| TP2: Experimental vs control | 9.4 | < 0.001 |
| Experimental: TP1 vs TP2 | 7.9 | < 0.001 |
| Control: TP1 vs TP2 | 1.9 | ns |

in Step 2 improved model fit, $\Delta F(1,80) = 52.33$, $p < .001$, $\Delta R^2 = .27$, increasing the explained variance to 52.5% ($R^2 = .55$, adj. $R^2 = .54$). Engagement introduced in Step 3 did not significantly increase the explained variance. In Step 4, education level added a small but significant increment, $\Delta F(1,78) = 5.88$, $p = .018$, $\Delta R^2 = .03$) and engagement—introduced in the previous step—became significant in this model. In Step 5 prior parenting experience did not significantly improve the model, $\Delta F(1,77) = 2.82$, $p = .097$, $\Delta R^2 = .01$. The final model explained 60.8% of variance in TP2 score ($R^2 = .61$, adj. $R^2 = .58$), $F(5,77) = 23.85$, $p < .001$ (see Table 6 for model summary).

In the final model, four predictors were significant: group assignment, baseline score at pretest, engagement, and education (see Fig 2). Participants in the control group scored lower at TP2 than those in the experimental group (B = −8.13, SE = 1.32, $t = -6.18$, $p < .001$). Higher TP1 scores predicted higher TP2 scores (B = 0.53, SE = 0.09, $t = 6.08$, $p < .001$). Engagement was positively associated with TP2 score (B = 0.23, SE = 0.10, $t = 2.38$, $p = .020$), and higher level of education also predicted higher TP2 scores (B = 2.36, SE = 1.07, $t = 2.20$, $p = .031$). Prior parenting experience (having older children) was not a significant predictor (B = −2.63, SE = 1.56, $t = -1.68$, $p = .097$).

**Table 6. Model summaries for the hierarchical linear regression analysis of predictors of parental beliefs' alignment with scientific evidence after the intervention (non-preregistered regression analysis 1).**

| Step/ Predictor | B | SE | t | $R^2$ | $\Delta R^2$ |
|---|---|---|---|---|---|
| **Step 1** | | | | .29 | – |
| (Intercept) | 52.73*** | 1.10 | 48.03 | | |
| Group (2) | −9.62*** | 1.69 | −5.69 | | |
| **Step 2** | | | | .55 | .27*** |
| (Intercept) | 29.28*** | 3.51 | 8.35 | | |
| Group (2) | −7.78*** | 1.37 | −5.67 | | |
| Pretest score (TP1) | 0.52*** | 0.08 | 6.90 | | |
| **Step 3** | | | | .56 | .01 |
| (Intercept) | 27.87*** | 3.63 | 7.69 | | |
| Group (2) | −7.87*** | 1.37 | −5.77 | | |
| Pretest score (TP1) | 0.53*** | 0.08 | 7.00 | | |
| Engagement | 0.14 | 0.10 | 1.43 | | |
| **Step 4** | | | | .59 | .03* |
| (Intercept) | 17.99** | 5.42 | 3.32 | | |
| Group (2) | −8.15*** | 1.33 | −6.12 | | |
| Pretest score (TP1) | 0.46*** | 0.08 | 5.89 | | |
| Engagement | 0.22* | 0.10 | 2.24 | | |
| Education | 2.58* | 1.08 | 2.40 | | |
| **Step 5** | | | | .61 | .01 |
| (Intercept) | 16.85** | 5.40 | 3.12 | | |
| Group (2) | −8.13*** | 1.32 | −6.18 | | |
| Pretest score (TP1) | 0.53*** | 0.09 | 6.08 | | |
| Engagement | 0.23* | 0.10 | 2.38 | | |
| Education | 2.36* | 1.07 | 2.20 | | |
| Has Children (Yes) | −2.63 | 1.56 | −1.68 | | |

**Note**. *** $p < .001$, ** $p < .01$, * $p < .05$.

Supplementary bivariate correlations in the full sample ($N = 84$) showed that the TP1 score was positively associated with education, $r(82) = .31$, $p = .004$, and with the binary indicator of prior parenting experience, $r(82) = .44$, $p < .001$, but was not related to engagement, $r(82) = −.05$, $p = .656$.

**Predictors of the increase in parental beliefs alignment with science: Preregistered regression analysis 2**

A hierarchical linear regression was conducted to identify predictors of the increase in parents' alignment of beliefs with scientific evidence between the pretest (TP1) and posttest (TP2) (i.e., the gain resulting from taking part in the intervention). The outcome variable was the increase in score from TP1 to TP2 (the gain). Predictors were entered one by one in the following order: (1) baseline score at pretest (TP1), (2) engagement during the workshops; followed by the demographic variables: (3) education level and (4) prior parenting experience (having older children in the household: yes/no).

We included all participants who took part in the experimental (language) intervention—both those assigned to the randomized experimental group and those in the additional non-randomized group—provided that they completed both the pretest and posttest questionnaires. In total, **48 participants** were included in this analysis (**n = 40** from the randomized group and **n = 8** from the non-randomized group; see Table 7 for group characteristics).

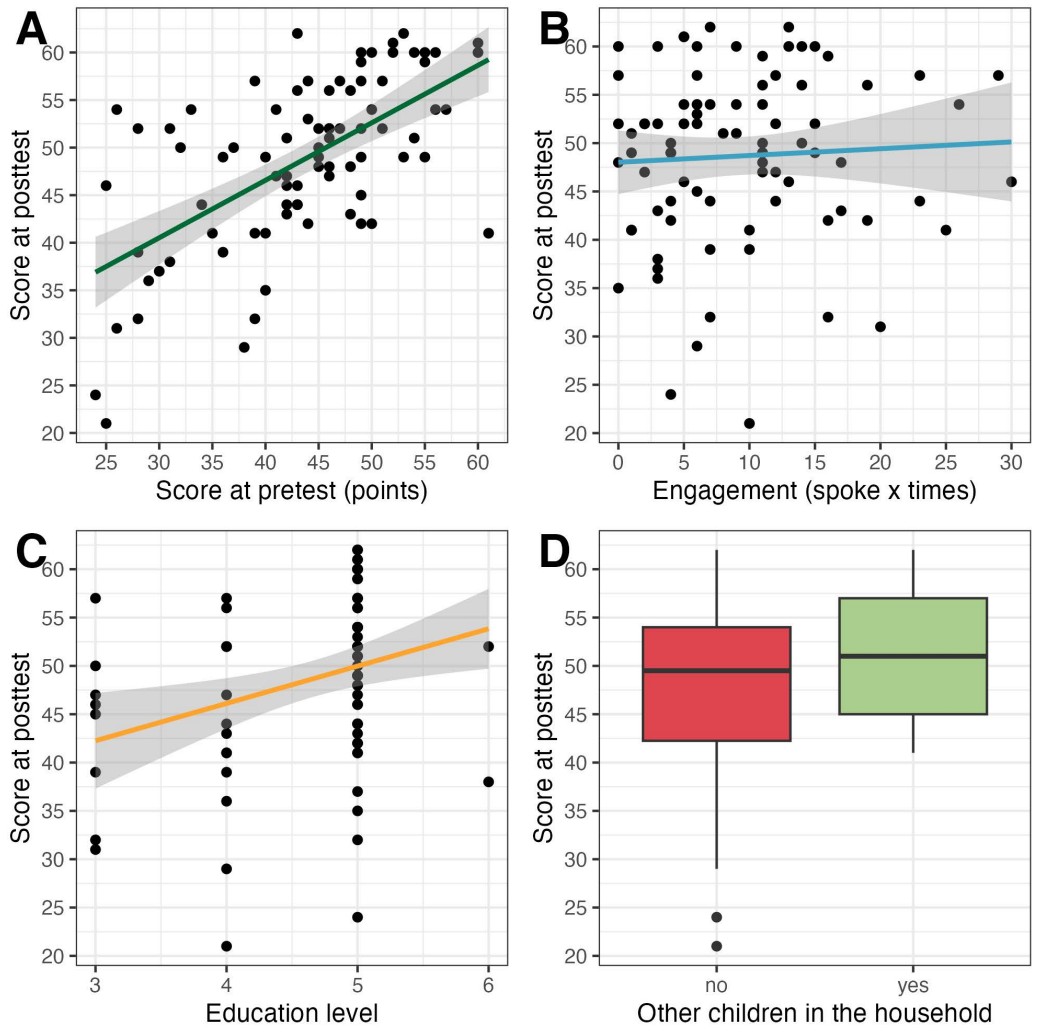

**Fig 2. Predictors of participants' beliefs alignment with scientific evidence after taking part in the experimental or control intervention with: (A) the level of alignment of participants' beliefs with the scientific evidence before the intervention, (B) the level of engagement in the intervention, (C) participants' education, and (D) prior parenting experience.** Note. In A, C, and D n = 84; in B n = 83.

In Step 1, the model including the pretest score was significant and explained 51.3% of the variance in the increase in score ($R^2 = .51$, adj. $R^2 = .50$), $F(1,46) = 48.45$, $p < .001$. The addition of engagement (Step 2), education (Step 3), and prior parenting experience (Step 4) did not improve model fit (see Table 8 for detailed statistics and summaries of each model). In the final model, lower TP1 score predicted larger gains from the intervention (B = −0.59, SE = 0.11, $t = −5.24$, $p < .001$), indicating that participants who started with less scientifically aligned beliefs benefited more. Engagement (B = 0.24, SE = 0.13, $t = 1.94$, $p = .060$), education (B = 1.80, SE = 1.43, $t = 1.26$, $p = .214$) and prior parenting experience (B = −3.36, SE = 2.07, $t = −1.63$, $p = .111$) were not significant predictors (see Fig 3).

Supplementary correlations within the language intervention subgroup ($N = 48$) indicated that higher TP1 score was associated with prior parenting experience ($r(46) = .51$, $p < .001$), but it was not associated with engagement ($r(46) = −.13$, $p = .364$). The correlation of TP1 score with education approached significance ($r(46) = .28$, $p = .051$).

**Table 7. Characteristics of participants included in the analysis of the predictors of the intervention effectiveness (preregistered regression analysis 2).**

| Baseline characteristic | N |
|---|---|
| **Total** | 48 |
| **Gender** | |
| Female | 44 |
| Male | 4 |
| **Prior parenting experience** | |
| Expecting first child | 32 |
| Rising older child(ren) | 16 |
| **Highest educational level** | |
| Secondary school | 4 |
| Incomplete higher education | 9 |
| Tertiary education – BSc; BA; Eng.; MSc; MA | 34 |
| Tertiary education – PhD | 1 |
| **Child's expected sex** | |
| Male | 18 |
| Female | 14 |
| Not known | 16 |
| **Questionnaire score** | |
| TP1 Pretest mean (SD) | 44.98 (9.15) |
| TP2 Posttest mean (SD) | 52.73 (6.68) |

**Note.** The mean age of participants was 32.85 years ($SD = 3.97$)

**Table 8. Model Summaries for hierarchical regression analysis of predictors of the increase in correlaparental beliefs alignment with science (preregistered regression analysis 2).**

| Step/ Predictor | B | SE | t | R² | ΔR² |
|---|---|---|---|---|---|
| **Step 1** | | | | .51 | – |
| (Intercept) | 37.74*** | 4.39 | 8.59 | | |
| Pretest score (TP1) | −0.67*** | 0.10 | −6.96 | | |
| **Step 2** | | | | .54 | .022 |
| (Intercept) | 35.28*** | 4.65 | 7.59 | | |
| Pretest score (TP1) | −0.65*** | 0.10 | −6.79 | | |
| Engagement | 0.18 | 0.12 | 1.47 | | |
| **Step 3** | | | | .56 | .022 |
| (Intercept) | 26.39** | 7.56 | 3.49 | | |
| Pretest score (TP1) | −0.68*** | 0.10 | −7.03 | | |
| Engagement | 0.25 | 0.13 | 1.92 | | |
| Education | 2.13 | 1.44 | 1.48 | | |
| **Step 4** | | | | .58 | .026 |
| (Intercept) | 24.77** | 7.49 | 3.31 | | |
| Pretest score (TP1) | −0.59*** | 0.11 | −5.24 | | |
| Engagement | 0.24 | 0.13 | 1.94 | | |
| Education | 1.80 | 1.426 | 1.26 | | |
| Has Children (Yes) | −3.36 | 2.07 | −1.63 | | |

**Note.** *** p < .001, ** p < .01, * p < .05.

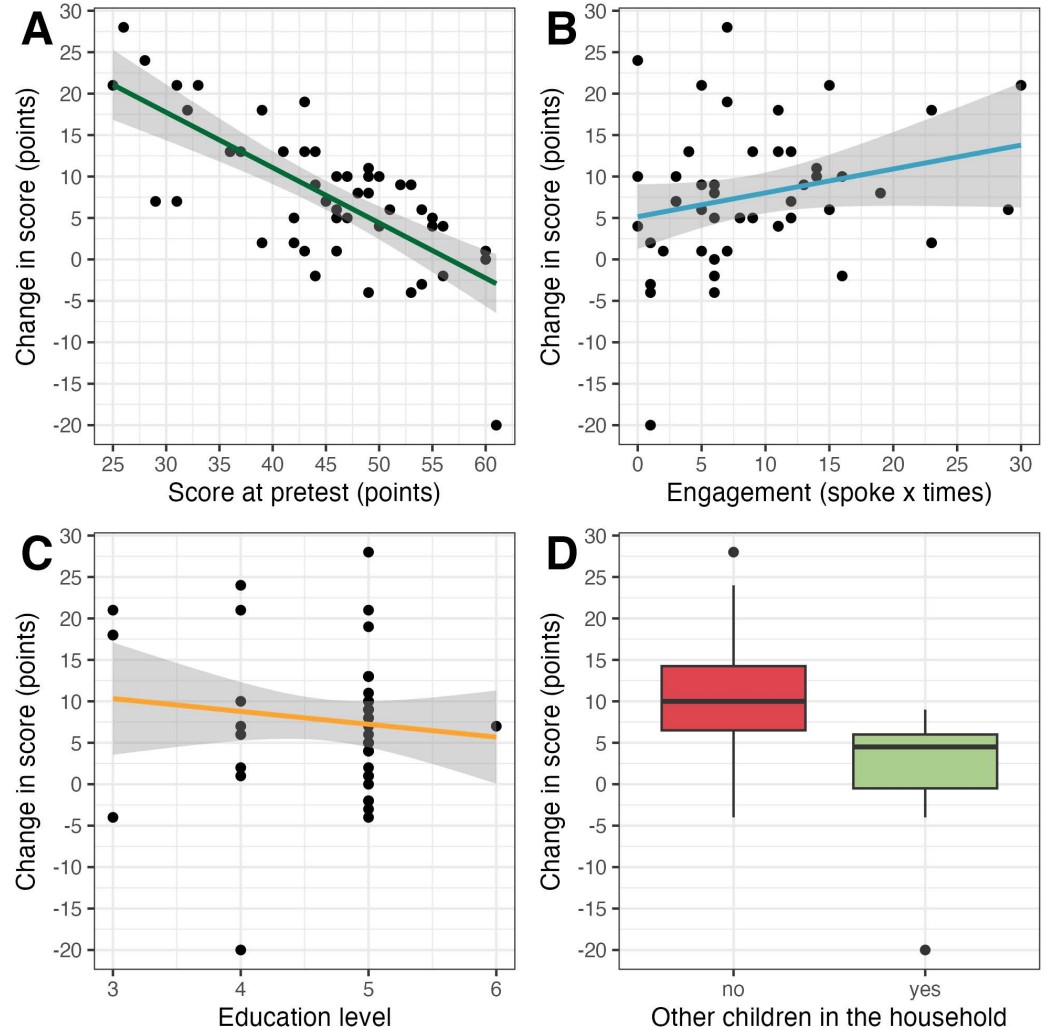

**Fig 3. Predictors of intervention effectiveness.** The relationship between the gain from the intervention and (A) the level of alignment of participants' beliefs with the scientific evidence before the intervention, (B) the level of engagement in the intervention (C) participants' education, and (D) prior parenting experience with whiskers extending to a maximum of 1.5 times the interquartile range. **Note.** In (A), (C) and (D) n = 48; in (B) n = 47. The y-axis represents the change in test score between TP1 and TP2, i.e., the difference between the posttest score and the pre-test score. For some participants an increase in score was observed (positive values) and for others a decrease (negative values).

## Maintenance of the intervention effects: Non-preregistered ANOVA analysis 2

The data from participants who were assigned to groups randomly and filled in the questionnaire at all three time points: pretest, posttest and long-term maintenance test (when the children were 9 months-old) were included in this analysis (N = 42, n = 17 in the experimental group, and, n = 25 in the control group). A mixed ANOVA was conducted to test whether the effect of the intervention persisted over time. The effect of the between-subject variable: group (the type of the workshop: experimental or control) on the alignment of parental beliefs was significant ($F(1,40) = 7.28$, $p = .010$, generalized $\eta^2 = .12$, medium effect size), indicating that beliefs of parents from the experimental group were more aligned with scientific evidence. Bonferroni post-hoc tests revealed that the groups did not differ significantly in the level of alignment of participants' beliefs with scientific evidence on language at pretest, but they differed at posttest (mean difference = 7.6,

$p$ = .009) and maintenance test (mean difference = 6.8, $p$ = .04) with beliefs in the experimental group being significantly more aligned with scientific advice on language than those of participants in the control group (see Fig 4). Table 9 presents descriptive statistics of questionnaire scores on parental beliefs about language development and bilingualism across the two groups and three timepoints, and Table 10 and Table 11 present the model summary and Bonferroni post-hoc's for this model, respectively.

The effect of the within-subject variable: time (pretest vs posttest vs long-term maintenance point) on parental beliefs was also significant ($F(2,80) = 16.39$, $p < .001$, generalized $\eta^2 = .09$, medium effect size), indicating that the alignment of parental beliefs with scientific evidence changed over time. Bonferroni post-hoc tests showed that this effect was a result of the difference between the participants' outcomes at the pretest and in the maintenance test (mean difference = 5.6, $p$ = .010). The difference between the participants' outcome in the posttest and the maintenance test was not significant, indicating that the gain from the intervention persisted after an extended period. The further investigation by Bonferroni

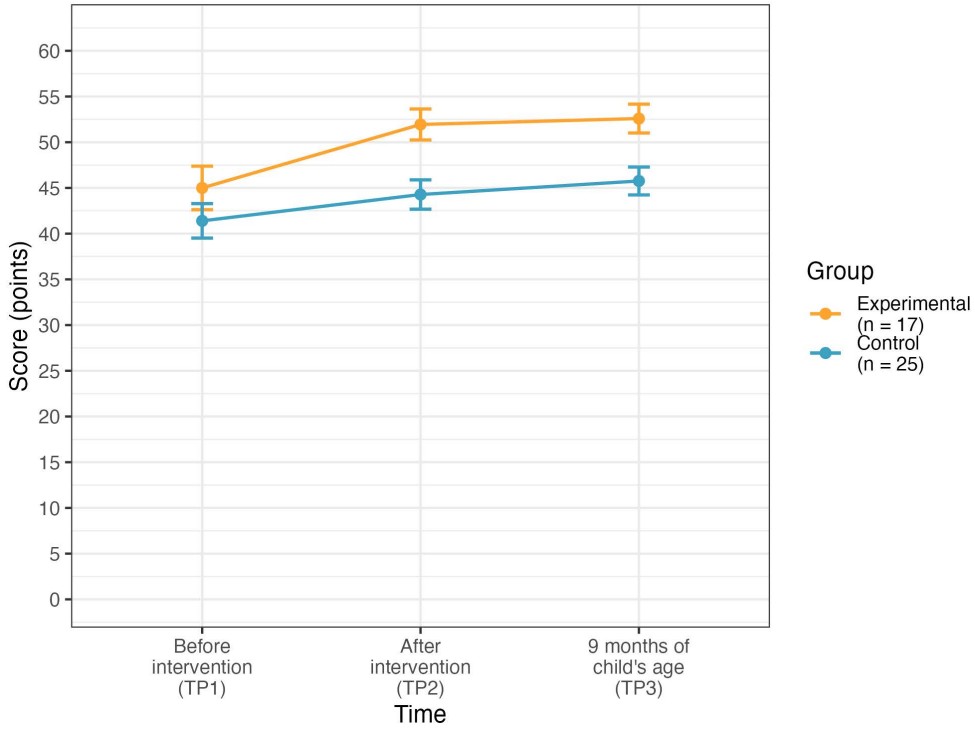

**Fig 4. Participants' performance at the pretest (TP1) and two posttests (TP2, TP3) in the experimental and control group.** Error bars represent standard errors of the mean (±SE).

**Table 9. Descriptive statistics for questionnaire scores on parental beliefs about language development and bilingualism across both groups and the three timepoints.**

| | TP1 | | TP2 | | TP3 | |
|---|---|---|---|---|---|---|
| **Group** | **M** | **SD** | **M** | **SD** | **M** | **SD** |
| Experimental – language workshop *n* = 17 | 45.0 | 9.83 | 51.9 | 6.99 | 52.6 | 6.50 |
| Control – sleep workshop *n* = 25 | 41.4 | 9.43 | 44.3 | 8.02 | 45.8 | 7.62 |

**Table 10. Non-preregistered mixed ANOVA analysis 2: results assessing the maintenance of intervention effects.**

| Effect | F | df1 | df2 | p | η2 |
|---|---|---|---|---|---|
| Group | 7.28 | 1 | 40 | .010 | .12 |
| Time point | 16.39 | 2 | 80 | <.001 | .09 |
| Group x Time point | 1.86 | 2 | 80 | .163 | .01 |

**Table 11. Bonferroni post-hoc test results for the maintenance of intervention effects.**

| Group comparison | Mean difference | p |
|---|---|---|
| TP1: Experimental vs control | 3.6 | ns |
| TP2: Experimental vs control | 7.6 | 0.009 |
| TP3: Experimental vs control | 6.9 | 0.04 |
| Experimental: TP1 vs TP2 | 6.9 | 0.041 |
| Experimental: TP2 vs TP3 | 0.7 | ns |
| Experimental: TP1 vs TP3 | 7.6 | 0.022 |
| Control: TP1 vs TP2 | 2.9 | ns |
| Control: TP2 vs TP3 | 1.5 | ns |
| Control: TP1 vs TP3 | 4.4 | ns |

post-hoc tests analysed per group revealed that the alignment of beliefs with scientific evidence on language of participants from the experimental group increased between the pretest and the posttest (mean difference=6.9, $p$=.041) and between pretest and the maintenance test (mean difference=7.6, $p$=.022). There were no significant differences between the posttest and the maintenance test. For the control group, no significant differences were observed between any time points.

Hence, the groups did not differ significantly in their initial level of the alignment of participants' beliefs with the scientific evidence (at pretest); but after the intervention (at the posttest), the beliefs in the experimental group were significantly more aligned with scientific knowledge than in the control group (mean difference=7.6, $p$=.002). Similarly, in the long-term maintenance test participants from the experimental group scored higher than participants from the control group (mean difference=6.8, $p$=.004) The interaction effect between group and time on parental beliefs was not significant.

## Discussion

Here, we evaluate the effectiveness of the first preregistered randomised controlled trial of an intervention for parents planning to raise their children in a bilingual environment. The intervention was delivered in Polish to Polish migrant parents in Norway.

Our findings indicate that an evidence-based online workshop on language development with a focus on bi-/multilingual development, when provided before or shortly after a child's birth, enhances parental beliefs' alignment with scientific evidence. Similar findings of gain in the alignment of parental beliefs with scientific evidence soon after a relatively short intervention directed to parents have been reported previously [15,29,30]. However, in contrast to previous findings [54], the effects of our intervention were sustained over time, remaining detectable nine months after the child's birth. To our knowledge, this study provides the first evidence that a parental language-focused intervention can produce such long-lasting changes in parents' knowledge or beliefs.

As opposed to earlier research, our population was not underprivileged and was not limited to families with children already exhibiting language problems or at risk for language disorders. This suggests that providing courses in language

development to the general public, or at least to parents expecting to raise their children bi-/multilingually, might improve the scientific base of these parents' beliefs [43]. However, our sample was characterised by two particular characteristics that may have positively influenced the intervention gain. Firstly, the study involved volunteers presumably interested in language development or in child development in general. Secondly, the respondents who registered for the study were willing to contribute to research, suggesting that they may have believed in evidence-based practices more than the average parent in that population. Thus, the characteristics of the sample may to some extent limit the generalisability of our findings. However, these kinds of participant characteristics are often present in intervention studies targeted at parents. Furthermore, the increase in alignment of parental beliefs with scientific evidence was observed only in the experimental group, indicating that the gains were specifically attributable to the intervention rather than to placebo effects or gradual shifts in parental beliefs that naturally occur as they gain experience raising an infant in a bilingual environment (preregistered ANOVA analysis 1).

Four participant characteristics were associated with the alignment of beliefs with scientific evidence following participation in either the experimental or control intervention: (a) the type of intervention to which the participant was assigned, (b) the level of scientifically based beliefs at study entry (i.e., the score obtained on the beliefs questionnaire at TP1, prior to the intervention), (c) the level of individual engagement in the intervention (measured as the frequency of spontaneous verbal contributions during the workshop), and (d) the participants' level of education. The fact that participation in the experimental workshop contributed more to having beliefs aligned with science at the end of study, than participation in the control intervention, confirmed the effectiveness of the intervention. Baseline alignment with science score significantly predicted post-intervention beliefs as well, indicating that participants' initial beliefs remained a strong determinant of subsequent scores. This finding is consistent with prior evidence demonstrating substantial temporal stability in knowledge-related constructs [55]. Although individual engagement did not account for additional variance, its predictive role became evident once demographic factors—particularly education—were taken into account. This pattern suggests that engagement and education share variance, and that controlling for educational background clarifies the unique contribution of engagement to post-intervention outcomes, so the effect of engagement may be obscured by educational attainment. Individuals with higher levels of education typically have stronger cognitive and metacognitive resources that facilitate the processing of workshop content: they more easily grasp abstract concepts related to early language development, integrate new information with existing knowledge, and navigate educational materials more efficiently [56]. As a result, their post-intervention scores are likely to be high regardless of how actively they participate during the workshop. In contrast, for participants with lower levels of education, active engagement may play a more critical role [57], as it helps compensate for a less extensive prior knowledge base and potentially greater difficulty in assimilating new material. This means that the effect of engagement on post-intervention beliefs becomes visible only after statistically accounting for education, which otherwise accounts for a substantial portion of the variance shared between both predictors. Although engagement and education remain relevant predictors of participants' beliefs after the intervention, the proportion of variance explained by these two factors is relatively small. Finally, prior parenting experience was the only factor among the predictors included in the model that was not significantly associated with post-intervention alignment of participants' beliefs with scientific evidence. This effect is difficult to interpret. On the one hand, it is possible that participation in the intervention helped first-time parents align their beliefs with scientific evidence to a level comparable to that of more experienced parents. An alternative explanation is that prior parenting experience was moderately correlated with pre-existing beliefs (i.e., the score at TP1). Therefore, the effect of prior parenting experience may already have been accounted for when the TP1 score was introduced in the second step of the model.

The preregistered regression analysis 2 showed that the *increase* in the alignment of parental beliefs with science (**the gain from the intervention**) was explained solely by the baseline level of the alignment (TP1, pretest score), showing that the less aligned with science participants' beliefs were at the beginning of the study, the more they gained from taking part in the intervention. All predictors added in subsequent steps (engagement, education, and parental status) failed to

improve the model and remained nonsignificant in the final model. The lack of predictive power of individual engagement for gains from the intervention suggests that its significance in explaining the change in parental beliefs may be tied to pre-existing knowledge, rather than to the anticipated causal mechanism whereby greater engagement would translate into greater learning. However it should be noted that this analysis was conducted on a smaller sample compared to non-preregistered regression analysis 1, as it was restricted to participants in the experimental condition and did not include data from the control group. Consequently, the reduced sample size may have limited statistical power, potentially obscuring effects that would have been detectable in a larger sample.

Particularly noteworthy are the findings that education level did not predict gains from the intervention (preregistered regression analysis 2) and showed only limited predictive value for the alignment of participants' beliefs with science after the intervention (non-preregistered regression analysis 1). Most of the early parental language interventions available in the literature are addressed to low SES participants [27], usually defined as the low-educated ones. Our result may be due to the limited variation in the participants' education in our sample. All of our participants had at least a secondary school education, and we had no participants with a primary or vocational middle school education level. Most of our participants held a university degree. Therefore, low variability in parental education might have obscured a potential effect of education on parental gains from intervention.

Another possible explanation for the low predictive value of education for parental beliefs after the intervention (non-preregistered regression analysis 1) —and its lack of predictive power for gains from the intervention (preregistered regression analysis 2)—is that education was already positively correlated with parental beliefs at the start of the study (TP1, pretest). Because the pretest score was entered into the model before education and accounted for a substantial proportion of the variance, it may have obscured the effect of education level on intervention gains.

Contrary to our predictions, prior parenting experience did not predict either participants' post-intervention beliefs or their gains from the intervention. The reason for this might be that parents expecting their first child presented a slightly but significantly lower level of alignment with scientific evidence at the pretest compared to those who already had children. This may reflect fewer opportunities and less motivation to observe young children, ask questions about their development, and search for evidence-based guidelines. As a result, any advantage associated with parenting experience may already have been captured by participants' initial knowledge, entered in earlier steps of the model. Consistent with previous research [58], parenting knowledge varies with experiential and sociodemographic factors, with more experienced parents generally demonstrating higher domain-specific understanding. In this sense, prior experience did not exert an independent effect beyond baseline knowledge.

It should be noted that some degree of overlap between predictors was confirmed by supplementary correlation analyses. They indicated that TP1 scores were positively correlated with both education and prior parenting experience. This pattern suggests partial shared variance among these variables, which may have reduced the apparent unique contribution of education or prior experience in multivariable models. Thus, the non-significant effects of some predictors should be interpreted cautiously, as they may reflect overlapping informational resources rather than the absence of substantive influence. Although the correlations were moderate rather than high, they indicate that multicollinearity may have attenuated some individual regression coefficients.

Taken together, we propose that while targeting only at-risk groups could potentially lower costs by directing resources to those most in need, pre-birth or early interventions could possibly reduce the number of children with challenges, also within low-SES families, lowering the costs overall. Providing interventions to families prior to the birth of their children implies engaging with them before any potentially maladaptive behaviors have emerged, thereby reducing the risk of stigma or shame associated with the need for change. Furthermore, the absence of established maladaptive behaviors eliminates the necessity of replacing entrenched habits, a process typically regarded as challenging and effortful. The advantage of our intervention is that it was relatively short (4.5 hours in total) and delivered online. As such, it has the potential to be scaled up and integrated into the regular programme of pre-birth classes for parents-to-be.

## Limitations

The study has some limitations. First, the goal of the intervention was not only to increase parental knowledge of language development, but, most importantly, to improve the quality and quantity of language heard by children, and as a result, to support children's language development. In the current paper, we do not analyse data on children's language development. Such data is currently being collected from some of the parents, using the StarWords mobile application [59] and standardized MacArthur-Bates Communicative Development Inventories [Norwegian: 60, Polish: 61]. In the future, we hope to conduct analyses on the language development of children whose parents participated in the intervention. However, data collection in a longitudinal study is time-consuming as we must wait until the children grow up. Due to the high drop-out rate in the study, we expect to collect data for a smaller number of children than the number of parents included in the current analysis.

Another limitation is the relatively small size of the study sample (especially at TP3), due to the recruitment difficulties and drop-out rate at each stage of the study. Although the results from this study show promise, especially the part related to gain maintenance after several months, they should be treated with caution because of the smaller number of participants in this exploratory analysis.

Finally, the sample used in this study was relatively homogeneous, which may limit the generalizability of our findings. On the one hand, focusing on Polish-speaking participants living in Norway offers a clear advantage, as it enables a detailed examination and comparison of children's language development in both languages at later stages of the study. On the other hand, this raises questions about the extent to which our results are applicable to other populations – including Polish immigrants in different countries, and even more so to parents raising bilingual children in more diverse contexts – for example, in countries with multiple distinct official languages, such as Luxembourg, or among refugee families whose migration is driven by war rather than economic reasons.

Last but not least, our sample consisted mainly of well-educated Polish women living in Norway, with no participants reporting only primary education. While this group may be representative of young Polish female migrants in Norway, 47% of whom hold a university degree and only 14% have a high school education or lower [62], this demographic homogeneity may limit the applicability of our findings to more diverse populations, including less-educated parents or those in different socio-cultural contexts.

## Conclusion

This is the first randomised controlled trial assessing the effectiveness of the intervention for migrant parents raising their infants bilingually, conducted on the population of Polish migrants to Norway. We were able to improve our participants' beliefs on early language development (and bilingual development in particular) alignment with scientific evidence not only directly after delivering the intervention but also several months later.

We translated the intervention content and materials to English and Norwegian and the translations along with the Polish original are publicly available on the OSF (https://osf.io/46fvq/). We hope that this can lead to other studies, examining the efficiency of the intervention in more diverse settings.

## Acknowledgments

We would like to thank all the participating families for their longstanding involvement in the study. We are also grateful to our collaborators: Weronika Białek, Natalia Wojtkowiak and Ewa Komorowska who supported the organization of the workshops, Monika Stąpor and Sylwia Stasiak who helped in the recruitment process, Magdalena Komsta and Agata Woźnicka who helped to improve the form of the workshop and Karla Orban and Jagoda Sikora who facilitated workshops. We would also like to thank research assistants in Norway, who supported the recruitment for the study. The materials for the intervention were translated into English and Norwegian and published with support from the bilateral initiative

"Science & Society: Bilateral Initiative in the Social Sciences, Arts and Humanities", funded by the Norway and EEA Grants 2014–2021 (agreement No. 2024/43/7/HS6/00002) and awarded to Jagiellonian University, partner: University of Warsaw /activity 11. We acknowledge the use of ChatGPT and DeepL to support proofreading. All generated suggestions were manually reviewed, and the authors take full responsibility for the final version of the manuscript.

## Author contributions

**Conceptualization:** Agnieszka Dynak, Katarzyna Bajkowska, Magdalena Krysztofiak, Magdalena Łuniewska, Karolina Muszyńska, Ewa Haman.

**Data curation:** Agnieszka Dynak, Katarzyna Bajkowska.

**Formal analysis:** Agnieszka Dynak, Katarzyna Bajkowska.

**Funding acquisition:** Agnieszka Dynak, Magdalena Łuniewska, Nina Gram Garmann, Ewa Haman.

**Investigation:** Agnieszka Dynak, Katarzyna Bajkowska, Jolanta Kilanowska, Magdalena Łuniewska.

**Methodology:** Agnieszka Dynak, Katarzyna Bajkowska, Jolanta Kilanowska, Joanna Kolak, Magdalena Krysztofiak, Magdalena Łuniewska, Karolina Muszyńska, Nina Gram Garmann, Ewa Haman.

**Project administration:** Nina Gram Garmann, Ewa Haman.

**Resources:** Agnieszka Dynak, Katarzyna Bajkowska, Magdalena Krysztofiak, Magdalena Łuniewska, Karolina Muszyńska, Ewa Haman.

**Supervision:** Agnieszka Dynak, Nina Gram Garmann, Ewa Haman.

**Visualization:** Agnieszka Dynak, Katarzyna Bajkowska.

**Writing – original draft:** Agnieszka Dynak, Katarzyna Bajkowska, Magdalena Łuniewska, Nina Gram Garmann.

**Writing – review & editing:** Agnieszka Dynak, Katarzyna Bajkowska, Jolanta Kilanowska, Joanna Kolak, Magdalena Krysztofiak, Magdalena Łuniewska, Karolina Muszyńska, Nina Gram Garmann, Ewa Haman.

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
