## [Decision Letter · Decision Letter 0]

29 Jan 2025

Dear Dr. Bajkowska,

We look forward to receiving your revised manuscript.

Kind regards,

Mary Diane Clark, PhD

Academic Editor

PLOS ONE

“This study is one part of the PolkaNorski project which was funded by the Norwegian Financial Mechanism 2014–2021 within the NCN GRIEG programme (UMO-2019/34/H/HS6/00615).”

Additional Editor Comments:

Reviewer 1 was not truly able to adequately review this paper. However, Reviewer 2 provides specific input about increasing the impact of your study. Therefore, please take those suggestions to guide your revision. Thank you for submitting to Plos One. I look forward to your revision.

Best

Reviewers' comments:

Reviewer's Responses to Questions

**Comments to the Author**

1. Is the manuscript technically sound, and do the data support the conclusions?

Reviewer #1: No

Reviewer #2: Yes

2. Has the statistical analysis been performed appropriately and rigorously?

Reviewer #1: I Don't Know

Reviewer #2: Yes

3. Have the authors made all data underlying the findings in their manuscript fully available?

Reviewer #1: Yes

Reviewer #2: Yes

4. Is the manuscript presented in an intelligible fashion and written in standard English?

Reviewer #1: Yes

Reviewer #2: Yes

Reviewer #1: Only the experimental group received the intervention on dialogic reading impt, language miletones recog.etc. The control group got a workshop only on sleep patterns so of course, they did not perform on language issues they were not trained in. It would make more sense to have a different language intervention for the control group. They didn't get any information on the issues they were evaluated on. I may be missing something, but this study's outcomes made no sense to me.

Reviewer #2: PONE-D-24-46478

Thank you for the opportunity to review this interesting work Better start to bilingual development: Bridging parental beliefs and science through early intervention

The study evaluates an intervention designed to align parental beliefs about language

development and bilingualism with science-based recommendations in a RTC study based on 76 polish parents living in Norway. The study showed that the intervention (in the form of three workshops) increased alignment of parental beliefs post intervention and 9 months after the intervention this increase was maintain. Some moderation effects were established so that those with the least aligned beliefs at posttest, first -time parents and highly engaged parents gained the most. The trial was pre-registered.

The study has several strengths not least the RCT design with both a post measure and a nine-months follow-up measure and moderation analysis that shed some light of who benefited most from the intervention. Moreover, the sample is not restricted to low-SES parents with children with language delay as is often the case. Even though no behavioral data to support that change in beliefs are related to parental behavioral changes or associations between change and child outcome data are availbale this is a strong first step which add to the current database.There are some aspects of this manuscript that could be bolstered to make the draft stronger and more convincing.

The theoretical background is well laid out in the introduction. There are places, though, where it is difficult to distinguish between types of interventions referred to (behavioral interventions or belief interventions).

The structure and design of the intervention is well described, discussion and limitation of findings as well. It is mainly in the method section that is unclear.

Recruitment and randomization procedures are well described but the method section would benefit from a clearer organization. First of all, a section devoted to measures would be useful. Here we need more information about the content in the researcher developed measures of parent beliefs with examples of items, as well as its psychometric properties. Did the authors run exploratory factor analysis to confirm that all items load on the same factor or is the questionnaire tapping into different constructs relating to parental beliefs? If more than one, what are scores on different constructs and correlations? What is the internal consistency of the scale? Were any of the co-variates related to this index at pretest?

Secondly, more information about the engagement coding would be useful for interpreting these results. I assume that the engagement measure is tapping into (an aspects of) parents’ fidelity of the intervention, but what type of behaviors were coded and how. However, it would be useful with information about how many parents completed all workshops. If there is any variation here, this could be an interesting predictor too.

There is some mentioning of the analytic strategy in the section Parent questionnaire and variables measured and in the Results section but I think this information is better placed in an analytic strategy section. In that section, more information about choice of model would be good. Both the mixed ANOVA and the linear regression used in the moderation analysis should be described in the Analytic section (it is btw not clear to me why the non-randomized language workshop group is included in the moderation analysis (not least as the effect measures are different). Also, exact description of attrition is lacking (and n is not reported in tables). Judge from plots there is a high attrition rate between T2 and T3 but how was the missingness handled in the analyses. I also lack information about if and which co-variates is used as control variables in the main model and the moderation model.

Limitations are generally well addressed. The fact that it is a relatively small sample size highly skewed towards higher educated parents may account for the fact that SES did not predict change as also mentioned by authors. but I wonder how the fact that all participants come from one language background affect the generalizability of the results. It would be good to hear the authors reflections on this.

.

Reviewer #1: No

Reviewer #2: No

---

## [Author Response · Author response to Decision Letter 1]

22 Aug 2025

Thank you for your work and insightful comments. We provide a point-by-point response to each of the 2nd Reviewer’s comments in the attached file, indicating changes we made in our manuscript or providing a suitable rebuttal. To facilitate the review, changes in the manuscript are marked in blue. We hope that our replies and the changes added to our manuscript satisfy the your concerns, and have significantly improved the quality of our paper. The text was also proofread and some minor stylistic changes were introduced.

---

## [Decision Letter · Decision Letter 1]

22 Sep 2025

Dear Dr. Bajkowska,

Thank you for all of the changes that you made for the resubmission--it has made the paper much stronger.  I have identified some smaller issues that need to be addressed. It seems to me that there are some mistakes between the results and discussion. Also, the analyses need tables.  The two ANOVAs need ANOVA tables and the regression needs a table also but it also needs more description of how you entered the predictors.  I would like to see R change scores to see how much additional variance is explained by other significant predictors.  It would be better to have the t tests for the significant predictors instead of the betas.  If you add the F tests showing the significance of the overall model and then the t tests for the significant predictors with their R square change scores you will be able to describe in more detail the important variables.

These things are given in more detail in the attached file.

We look forward to receiving your revised manuscript.

Kind regards,

Mary Diane Clark, PhD

Academic Editor

PLOS ONE

Journal Requirements:

Additional Editor Comments:

Thank you for all of the changes that you made for the resubmission--it has made the paper much stronger. I have identified some smaller issues that need to be addressed. It seems to me that there are some mistakes between the results and discussion. Also, the analyses need tables. The two ANOVAs need ANOVA tables and the regression needs a table also but it also needs more description of how you entered the predictors. I would like to see R change scores to see how much additional variance is explained by other significant predictors.

These things are given in more detail in the attached file.

Reviewers' comments:

Reviewer's Responses to Questions

**Comments to the Author**

Reviewer #1: All comments have been addressed

2. Is the manuscript technically sound, and do the data support the conclusions?

Reviewer #1: Yes

3. Has the statistical analysis been performed appropriately and rigorously?

Reviewer #1: Yes

4. Have the authors made all data underlying the findings in their manuscript fully available?

Reviewer #1: Yes

5. Is the manuscript presented in an intelligible fashion and written in standard English?

Reviewer #1: Yes

Reviewer #1: The authors have adequately addressed the concerns from previous editors. I recommend they review the article one last time for English cohesion and grammar.

.

Reviewer #1: No

While revising your submission, please upload your figure files to the Preflight Analysis and Conversion Engine (PACE) digital diagnostic tool, https://pacev2.apexcovantage.com/. PACE helps ensure that figures meet PLOS requirements. To use PACE, you must first register as a user. Registration is free. Then, login and navigate to the UPLOAD tab, where you will find detailed instructions on how to use the tool. If you encounter any issues or have any questions when using PACE, please email PLOS at . PACE helps ensure that figures meet PLOS requirements. To use PACE, you must first register as a user. Registration is free. Then, login and navigate to the UPLOAD tab, where you will find detailed instructions on how to use the tool. If you encounter any issues or have any questions when using PACE, please email PLOS at . PACE helps ensure that figures meet PLOS requirements. To use PACE, you must first register as a user. Registration is free. Then, login and navigate to the UPLOAD tab, where you will find detailed instructions on how to use the tool. If you encounter any issues or have any questions when using PACE, please email PLOS at . PACE helps ensure that figures meet PLOS requirements. To use PACE, you must first register as a user. Registration is free. Then, login and navigate to the UPLOAD tab, where you will find detailed instructions on how to use the tool. If you encounter any issues or have any questions when using PACE, please email PLOS at figures@plos.org. Please note that Supporting Information files do not need this step.. Please note that Supporting Information files do not need this step.

---

## [Author Response · Author response to Decision Letter 2]

28 Nov 2025

Thank you for your work and insightful comments. We provide a point-by-point response to each of the Editor's and 2nd Reviewer’s comments in the attached file, indicating changes we made in our manuscript or providing a suitable rebuttal. To facilitate the review, changes in the manuscript are marked in blue. We hope that our replies and the changes added to our manuscript satisfy the your concerns, and have significantly improved the quality of our paper. The text was also proofread and some minor stylistic changes were introduced.

---

## [Editor Report · Decision Letter 2]

22 Dec 2025

Dear Dr. Dynak,

We look forward to receiving your revised manuscript.

Kind regards,

Mary Diane Clark, PhD

Academic Editor

PLOS One

Journal Requirements:

Additional Editor Comments:

Thank you for the many changes on the paper. The results are much improved and clearer. There are still a few issues but overall that section is greatly improved. The Discussion needs a great deal of work as much of it now is repeating what the results found.

The use of ChatGPT has added a great deal of redundancy to the lit review. I have made many suggestions and I do not think that those changes will take much time.

If you complete these changes---then I can go ahead and accept the newest revision.

---

## [Author Response · Author response to Decision Letter 3]

6 Mar 2026

Thank you for all your comments and suggestions on how to further improve our manuscript; we very much appreciate them. We provide a point-by-point response, including the corresponding changes in the manuscript, in the attached file.

---

## [Editor Report · Decision Letter 3]

12 Mar 2026

Dear Dr. Dynak,

We look forward to receiving your revised manuscript.

Kind regards,

Mary Diane Clark, PhD

Academic Editor

PLOS One

Journal Requirements:

Additional Editor Comments:

This paper reads extremely well now. Thank you for the care with the edits. You are going to groan---PLOS ONE does not have an editorial staff and the paper is in PDF. There are 3 extremely minor issues that will honestly require 2 minutes to fix. they are below.

1--Line 151 page 5 & needs to be replaced with the word 'and'

2--line 180 page 8 there are two periods. .. please delete one

3--line 37 page 16 since refers to time and you have used that in a sentence that the semantic meaning is cause. Can you replace that with either 'as' or 'because'. (honestly I would not flag this without the others)

Sorry to keep this ongoing.

---

## [Author Response · Author response to Decision Letter 4]

13 Mar 2026

Thank you very much for your careful reading and for pointing out the mistakes that were missed during our work on the revised version. We have corrected all issues you indicated and hope that no further errors remain.

---

## [Editor Report · Decision Letter 4]

16 Mar 2026

Better start to bilingual development: Bridging parental beliefs and science through early intervention for Polish families living in Norway

PONE-D-24-46478R4

Dear Dr. Dynak,

We’re pleased to inform you that your manuscript has been judged scientifically suitable for publication and will be formally accepted for publication once it meets all outstanding technical requirements.

Kind regards,

Mary Diane Clark, PhD

Academic Editor

PLOS One

Additional Editor Comments (optional):

Thank you for your patience. I enjoyed working with you all.
---

## [Editor Report · Acceptance letter]

PONE-D-24-46478R4

PLOS One

Dear Dr. Dynak,

I'm pleased to inform you that your manuscript has been deemed suitable for publication in PLOS One. Congratulations! Your manuscript is now being handed over to our production team.

Kind regards,

on behalf of

Dr. Mary Diane Clark

Academic Editor

PLOS One